# Characterizing Epitope Binding Regions of Entire Antibody Panels by Combining Experimental and Computational Analysis of Antibody: Antigen Binding Competition

**DOI:** 10.3390/molecules25163659

**Published:** 2020-08-11

**Authors:** Benjamin D. Brooks, Adam Closmore, Juechen Yang, Michael Holland, Tina Cairns, Gary H. Cohen, Chris Bailey-Kellogg

**Affiliations:** 1Department of Biomedical Sciences, Rocky Vista University, Ivins, UT 84738, USA; 2Inovan Inc., Fargo, ND 58102, USA; 3Department of Microbiology, School of Dental Medicine, University of Pennsylvania, Philadelphia, PA 19104, USA; tmcairns@upenn.edu (T.C.); ghc@upenn.edu (G.H.C.); 4Department of Pharmacy, North Dakota State University, Fargo, ND 58102, USA; Adam.Closmore@gmail.com; 5Department of Biomedical Engineering, North Dakota State University, Fargo, ND 58102, USA; juechen.yang@gmail.com (J.Y.); mchollandred@gmail.com (M.H.); 6Computer Science Department, Dartmouth, Hanover, NH 03755, USA; cbk@cs.dartmouth.edu

**Keywords:** epitope binning, epitope mapping, epitope prediction, antibody:antigen interactions, protein docking, glycoprotein D (gD), herpes simplex virus fusion proteins

## Abstract

Vaccines and immunotherapies depend on the ability of antibodies to sensitively and specifically recognize particular antigens and specific epitopes on those antigens. As such, detailed characterization of antibody–antigen binding provides important information to guide development. Due to the time and expense required, high-resolution structural characterization techniques are typically used sparingly and late in a development process. Here, we show that antibody–antigen binding can be characterized early in a process for whole panels of antibodies by combining experimental and computational analyses of competition between monoclonal antibodies for binding to an antigen. Experimental “epitope binning” of monoclonal antibodies uses high-throughput surface plasmon resonance to reveal which antibodies compete, while a new complementary computational analysis that we call “dock binning” evaluates antibody–antigen docking models to identify why and where they might compete, in terms of possible binding sites on the antigen. Experimental and computational characterization of the identified antigenic hotspots then enables the refinement of the competitors and their associated epitope binding regions on the antigen. While not performed at atomic resolution, this approach allows for the group-level identification of functionally related monoclonal antibodies (i.e., communities) and identification of their general binding regions on the antigen. By leveraging extensive epitope characterization data that can be readily generated both experimentally and computationally, researchers can gain broad insights into the basis for antibody–antigen recognition in wide-ranging vaccine and immunotherapy discovery and development programs.

## 1. Introduction

The utility of antibodies (Abs) for treatment of disease has been recognized for over a century and the clinical realization of this vision in a host of broad therapeutic indications has begun to come to fruition [1,2]. The efficacy of an Ab, whether from vaccination or as a therapeutic, is functionally determined by the specific epitope that it recognizes on its cognate antigen (Ag). Thus, even Abs targeting the same Ag have demonstrated variable efficacy depending on their epitope specificities [3,4]. Recent advances in Ab engineering have enabled researchers to generate large panels of Abs with broad epitope coverage allowing for the selection of Ab–Ag pairings with improved safety and efficacy [5,6,7,8]. Abs have demonstrated differential effects in both vaccines and therapies depending on their epitopes [9,10,11]. Moreover, combination immunotherapies with diverse epitopes have demonstrated synergistic efficacy and have reduced the ability of cancer and infectious disease to develop resistance [12,13]. Furthermore, the use of immune repertoire B cell sequencing is supporting expanded clinical applications not only in immunotherapy, but also in the development of vaccines for emergent infectious diseases, both viral and bacterial [14]. The specificity of Abs in vaccine applications reveals various levels of protection depending on which epitopes are targeted by the immune response [12,13,15,16,17,18,19].

### 1.1. Understanding the Structures of Epitope:Paratope Interactions Guides the Design of Superior Immunotherapies and Vaccines

One of the areas for improvement in Ab engineering is the characterization of the binding interactions between an Ab and its specific Ag, as defined by the epitope–paratope interface [5,20,21,22]. Due to practical limitations, structural data is only available for a relatively small number of epitope–paratope interactions [23,24], since commonly used analytical techniques for obtaining these data, such as X-ray crystallography, NMR spectroscopy, cryo-electron microscopy, and H-D exchange mass spectrometry, are highly resource-intensive and often require artisan skillsets [25,26,27,28]. As such, these techniques are feasible only for late-stage lead molecules where they provide confirmation data rather than early-stage predictive tools that influence candidate selection [5]. These limitations highlight the realization that a more sophisticated strategy is required to characterize a panel of antibodies either to screen immunotherapies or to characterize immune repertoires from natural or vaccinated responses [23]. The localization of these interactions would provide information to help understand biochemical mechanisms of action, which is at the core of advancing the discovery and development of new immunotherapeutics and vaccines [20,29,30].

### 1.2. High-Throughput Epitope Characterization Assays Provide Valuable Epitopic Information

Epitope characterization has made significant progress in recent years as the throughput of biosensors has improved [5,31]. Foremost amongst emerging techniques for high-throughput epitope characterization is epitope binning, as it merges the speed and functional-site identification capabilities demanded by the biopharmaceutical industry [3,4,5,32]. Epitope binning is a competitive immunoassay where Abs are tested in a pairwise manner for their simultaneous binding to their specific Ag, thereby generating a blocking profile for each Ab showing how it blocks or does not block the others in the panel [32,33,34,35]. Abs with similar blocking profiles can be clustered together into a “bin”, or represented as a “community” in a network plot that illustrates the blocking relationships among the Abs. In an oversimplified generalization, Abs in the same community are assumed to recognize the same epitope region and generally block the binding of others in the community. Advances in throughput offered by emerging array-based, label-free methods are allowing epitope binning to be performed early in drug discovery [32,33,34,35]. This approach can enable the pipeline to be populated with Abs that are diverse epitopically and, consequently, functionally. A key limitation of epitope binning remains its inability to provide insights into the locations of the epitopes on the target [3,4,5,32]. 

### 1.3. Integrating High-Throughput Experiment and Computation Enables Characterization of Ab-Specific Epitopes

An emerging approach for localizing Ab recognition and characterizing Ab-specific epitopes involves coupling experimental data and computational modeling [9,36]. Purely computational approaches to epitope prediction are quick and inexpensive but are not yet of sufficient accuracy to be relied upon on their own [36,37,38,39,40]. However, the incorporation of even limited experimental data can help close the gap. For example, the EpiScope approach first constructs a homology model of an Ab based on its sequence, then computationally docks that model onto a structure or high-quality homology model of the Ag, and finally, designs focused mutagenesis experiments to test the docking models [9]. While it is generally not clear from computational scoring alone which docking model is the most accurate, it was shown in both retrospective and prospective studies that a small number (generally 3–5) of binding assays for computationally designed Ag variants can reliably enable the various docking models to be confirmed or rejected and thereby identify the general epitope region [9]. Complementarily, experimental data can be used to focus docking and energy minimization to better define binding mode or epitope [41]. In general, combined computational–experimental approaches balance cost and accuracy in characterizing epitope sites. 

Here, we take this general idea and scale it up from individual Abs to sets of Abs, presenting a new integrated experimental-computational approach (Figure 1) to characterize epitopes for an entire panel of Abs against an Ag by combining experimental binning with “dock binning,” a new computational counterpart based on analysis of docking models for all the Abs. With the application to a model system, glycoprotein D from herpes simplex virus, we show that this combination of powerful experimental and computational methods can help rapidly identify antigenic regions and localize Ab-specific epitopes. The approach promises to enable better understanding of Ab–Ag interactions at a larger scale, and ultimately improve to the design of vaccines and therapeutics. 

## 2. Results

We demonstrate the power of combined experimental and computational binning in application to an important target with a wealth of data available for our use: herpes simplex virus (HSV) glycoprotein D (gD). GD is a fusion protein found in HSV that has served as the standard by which all other HSV-2 vaccines are evaluated for safety and efficacy [42]. GD subunit vaccines have conferred some measure of protection against viral challenge [42,43,44], but gD subunit vaccines fail to prevent infection or latent infection [45]. GD serves as a particularly good target for demonstration of our approach due to the availability of a large panel of available Abs as well as variant Ag constructs that may be leveraged for subsequent analyses. The following sections decribe our process (Figure 1) to characterize the epitopes of a set of anti-gD Abs.

### 2.1. Experimental Binning Identified Four Communities of Anti-gD Abs

Step #1 of the workflow (following the schematic in Figure 1) is to perform experimental epitope binning on a panel of Abs, identifying communities of cross-competing Abs and selecting representatives from each community for further investigation. Figure 2 further overviews the general workflow for such epitope binning experiments. In previous studies, this general approach was applied specifically to gD. In particular, high-throughput SPRi technology in a classical sandwich assay format was used to assess competition between pairs of Abs, from a panel of 46 Abs, against four soluble Ag variants, gD from type 1 HSV truncated to the first 285 or 306 residues, and that from type 2 HSV likewise truncated to 285 or 306 residues [46,47,48]. Subsequent analyses presented here are all based on the 285 residue truncation of gD2. 

The studies showed that the Abs covered much of the surface of gD [4], with numerous Abs in each distinct community of cross-blocking Abs [46,47,48]. Six “sentinel” Abs, namely DL11, MC23, MC2, MC5, MC14, and 1D3, were selected as representatives (generally the most highly connected within each community) [49,50]. Of particular note, there exists a crystal structure of an additional Ab, E317 (PDB ID 3W9E) [51]. Thus, even though E317 is in the same community as the previously selected DL11, we included it here to serve as a structural “control”. These seven Abs represent the communities in serving as the subjects of the following computational and experimental analyses.

### 2.2. Computational Epitope Prediction Characterized the Putative Epitope Binding Regions

With the representative Abs selected, Step #2 (Figure 1) is to construct Ab homology models and dock them to the structure/model of the Ag. Ab homology modeling is generally very high quality (<1–2 Å level RMSD to native) for everything except the heavy chain CDR 3, which is more variable (more typically 3–6 Å, though it can be better) [52,53]. For the gD study, we used representative state-of-the-art methods, within Schrodinger BioLuminate (BioLuminate, Schrödinger, LLC, New York, NY, USA, 2020.) to perform this modeling, but note that many alternative approaches are available and may yield somewhat different Ab and Ab:Ag models. A homology model was constructed for each of the seven Abs; for control purposes, E317 was homology modeled on a different scaffold from its crystal structure (RMSD 1.43 Å). The crystal structure of the Ag gD2 was taken from PDB id 2C36 [54,55,56], with missing residues homology modeled. Docking models were then generated for each Ab model against the gD2 model using the Piper algorithm within Schrodinger BioLuminate. This yielded roughly thirty representative low-energy docking models per Ab [57]. 

As is common [9], the docking models were spread over much of the gD2 surface. In general, the quality of Ab:Ag complex models produced by docking has steadily improved [58], e.g., for 95% of the cases in one benchmark, a near-native model was within the top 30 models [40]. Thus, while we could not be confident in the accuracy of any particular model, we could hypothesize that, in aggregate, they included the antigenic sites, setting up the next stage in our analysis.

### 2.3. Dock Binning Grouped Models and Identified Broadly Antigenic Regions

In order to identify the “hottest” putative antigenic sites on the protein, worth experimentally probing across the Ab panel, Step #3 (Figure 1) performs dock binning and constructs an antigenic heatmap. In analogy to experimental binning, this step first characterizes competition among Abs for sites on the Ag, here according to the docking models. At this point in the analysis, since there are many (roughly 30 in our gD results) docking models for each Ab, and they may be widely dispersed over the Ag, competition between a pair of docking models from different Abs does not necessarily imply that the Abs themselves compete. However, it does imply that it would be informative to evaluate the binding of those Abs against the Ag sites for which the docking models compete (e.g., mutate such a site and experimentally assess changes in binding/competition). Such a test would provide, with a single experiment, information regarding both Abs’ binding sites. More generally, the more evidence there is for interaction with an Ag residue (the “hotter” that residue), the more generally and experimentally informative it should be to probe that residue for binding across the entire Ab panel. This insight is the basis for performing dock binning and constructing an associated antigenic heatmap.

The dock binning workflow (schematically illustrated in Figure 3) is relatively straightforward, mostly following that of experimental epitope binning (see again Figure 2). Now the competition heatmap is based on overlap in docking models, rather than experimental competition. A variety of different methods are possible for assessing this overlap; the results presented here are based on one that we call the “common interaction metric”, which considers two Ab docks that contact the same residue(s) to be competing. This score drives the clustering of the docks based on their patterns of competition. Here, the hierarchical clustering method from experimental epitope binning was used [34,35]. A community network map is then generated from the dendrogram.

Figure 4 and Figure 5 illustrate the results from applying this process general process to the Ab:gD2 docking models. Figure 4 shows the docking model clusters generated for the seven Abs against gD2 and Figure 5 their community relationships. Note that docks from each Ab are found across all communities. 

The last step in dock binning, where it can produce insights beyond those that possible with experimental binning alone, leverages the intuition introduced above: analyze the dock bins to identify common putative antigenic sites which may then be subjected to experimental probing. Here computationally identified noncovalent Ab–Ag interactions (hydrogen, electrostatic, pi, and van der Waals) were aggregated across all docks for all Abs to determine a score for each Ag residue in terms of its total number of interactions. Figure 6 (also the last column of Figure 4) shows the resulting “antigenic heatmap,” with residues of the protein structure colored according to the dock binning community (Figure 4 and Figure 5) and with darkness proportional to the aggregated scores. This antigenic heatmap highlights the “hot” antigenic residues whose experimental probing is likely to be most informative and can enable the efficient localization of epitopes of the whole Ab panel. 

### 2.4. Dock Binning Enabled Selection of Experimental Assays to Evaluate Antigenic Regions

At this point in the process (Step #4), critical Ag residues according to the antigenic heatmap are probed for Ab recognition (e.g., via point mutagenesis followed by a binding assay, to assess if a mutation away from native disrupts binding). In this study, we were fortunate that there already exists a wealth of existing data available regarding Ab–gD2 binding. We cross-referenced the “hot” residues from the antigenic heatmap against the available data (including Ab:gD2 variant binding, peptide binding, and known “monoclonal Ab resistant”, or MAR, mutations), considering the sentinel Abs as well as others in their communities (Appendix A) [46,48]. Figure 7 highlights the residues from the antigenic heatmap for which such binding data was available [59,60]. We consider these as the experimental evaluation of the hot residues from the antigenic heatmap. We note that, in other settings, mutations (individual or combination) could be computationally designed to evaluate the disruption of binding while preserving antigenic stability [9,61]. Thus, while we used a large number of experimental measurements in this study, we expect that a much smaller number of tests would suffice in practice, e.g., 3–5 variants sufficed to localize individual Ab epitopes in previous computationally directed studies [9]. 

### 2.5. Experimental Data Allowed Re-Docking to Focus Ab:Ag Models Based on Experimental Binding Data, thereby Localizing Each Ab’s Epitope Region

Finally (Step #5 from Figure 1), the experimental data is used to focus docking of each Ab against the Ag. Here, the experimental data (Figure 7/Appendix A) was used to focus docking toward (with “affinity”) residues confirmed to be important for an Ab’s binding and away from (with “repulsion”) those determined not to be. For example, DL11 was docked with an affinity towards the residues 213 and 218 and with repulsion away from the residues associated with Abs from other communities, since these Abs were determined from the initial experimental binning not to compete with DL11, and thus it is assumed the epitopes don’t overlap. 

The focused docks are then subjected to dock binning and antigenic heatmap construction as described for Step #3. Figure 8, Figure 9 and Figure 10 show the focused-docking results for the anti-gD Abs. In contrast to the initial unconstrained docking, docking models for an Ab are now more concentrated, focused on the experimentally important residues for the Ab. Consequently, the communities are now fairly homogeneous for each Ab, and the localization of each Ab on gD can be fairly well inferred from an associated hot region in the antigenic heatmap. For example, when compared to the crystal structure for our “structural control Ab” E317, 10 antigenic heatmap residues agree with those in the crystal structure, while 12 extend further out and four are missed (Appendix A). 

While there is no ground truth for the actual epitopes of the Abs other than E317, the agreement between the antigenic heatmap residues and those previously identified by various experiments (Figure 7/Appendix A) was quantified in terms of both centroid distances and common residues (table in the middle of Figure 10). Many of the distances are quite small, indicating that the dock binning region was centered in the same general region as the experimentally identified residues used to focus it. However, some of the regions have fairly few experimentally identified residues (e.g., red and blue communities), and the docking models typically expanded to cover a significantly larger region capturing more Ab–Ag surface complementarity. The quantification of agreement between residues in the antigenic heatmap and those experimentally used to focus the docking further illustrates that while docking largely stays in the focused region as intended, it does include some additional residues and omit some others. The disparity could indicate, for example, either that the dock binning missed some important residues contributing to recognition or that the experiments overestimated the importance of some residues. Likewise, either that dock binning found some additional residues that had not been discovered by previous experiments or that it was somewhat off-target. Such differences can then be the subject of further experimental investigation. 

## 3. Discussion

### 3.1. The Integration of Experimental and Computational Binning Provides Important Epitope Information That Can Inform Discovery

Here, we demonstrate the utility of combining computational modeling and high-throughput experimental data to characterize epitopes early in discovery and thereby enable more effective drug and vaccine development. Epitope binning can facilitate the identification of functional epitopes and is thus increasingly used as a primary or early second screen [5]. However, while experimental epitope binning can inform competition among Abs for an epitope, it cannot localize the epitope on the Ag [5]. In contrast, while computational methods can identify potential epitopes, they are currently limited in their accuracy [40]. Even in this study, we were confronted with a wide range of seemingly equally reasonable docking models, leading us to consider: (1) how to identify the most accurate dock, (2) how to identify highly immunogenic sites on the target, and (3) how to group the docks into epitope regions in order to compare with experimental binning experiments. We determined that identifying the most accurate dock (question 1) was not realistic based on the current accuracy of the computational methods. However, we realized that docks could help us identify putative highly antigenic sites (question 2). Furthermore, this information would be the next logical step in assessing experimental binning communities and would limit the possibilities to consider when attempting to map communities onto their epitopes (question 3). Thus, this combination of computational and experimental binning (Figure 1) can leverage the advantages of each method, complementarily, in order to localize Ab epitopes across an entire panel. 

The initial dock binning step provides not only putative antigenically hot regions but also important information toward the generation of mutants that can be used in epitope mapping and cross-antigen binning to test hypotheses regarding Ab–Ag binding and thereby localize epitopes. Dock binning identifies communities of related Ab docking models, while the antigenic heatmap summarizes possible epitope binding regions on the Ag surface. These two analyses together provide elegant, biologically-centered visualizations characterizing potential binding patterns for an entire Ab panel. Taking the epitope binding regions as hypothesized regions that are generally antigenic allows the design of targeted experiments to evaluate them. The underlying intuition for these experiments is that if an Ab truly recognizes a particular epitope, then a mutation to substantially “re-surface” that epitope should disrupt Ab binding, and thus evaluation of changes to binding can allow us to confirm or reject the epitope for the Ab. By testing a set of variants based on the epitope binding regions, we can thereby simultaneously test all the various hypotheses spanning the Ab panel and potential epitopes. 

The results that can be obtained from the dock binning approach are limited by how well the Ab models and the Ab:Ag docking models reflect reality, along with how effectively the selected mutational variants can reveal that reality. We used representative computational approaches here, and overall observed reasonable agreement between the results from our approach and the one available ground truth crystal structure. While the previous purely experimental studies reflect an extensive amount of effort, they also may not fully reflect reality to the same extent as a crystal structure does. Thus, we are limited in our ability to characterize general trends regarding how well docking-based approaches perform and why, and what their particular weaknesses may be. From some previous gD studies, however, we do know that two of the Abs (MC2 and MC5) recognize epitopes that are at least partially obscured until receptor binding allosterically drives a conformational change [46,59]. We thus hypothesize that the epitope region identified for MC2, and presumably that for all members of the community it represents, is not particularly accurate. In general, epitopes, as well as paratopes, in conformationally dynamic or poorly modeled regions may present challenges for this model-driven approach.

### 3.2. While Binning Is Still Limited, Further Computational Advances Will Improve Its Accuracy and Informativeness

Significant limitations still exist with the prediction of protein-protein interactions, in particular with Ab-Ag epitope identification. Numerous opportunities exist to improve the process and better support vaccine and drug candidate screening. We here summarize some interesting avenues for computational development, following along the path of our workflow (Figure 1):
Experimental epitope binning. The fundamental question here is how to cluster the Abs into communities. and new and emerging techniques from network/community analysis, especially in genomics [62,63,64], may prove beneficial. These metrics can be used to identify communities that require further refinement and/or changes to the clustering. Furthermore, the clustering need not be discrete, i.e., the partition into communities can be overlapping, and this information can potentially be leveraged throughout the rest of the process. Such analysis could be particularly important in cases of partial or aberrant competition. While the “sentinel” antibodies used here were selected as most representative of clear-cut communities, future approaches could include algorithmic selection of representatives based on properties of the communities (e.g., ensuring adequate coverage of ambiguous clusters) and of the individual antibodies (e.g., based on sequence and structural analysis, favorability for development, etc.).Ab modeling and Ab–Ag docking. We used representative high-quality Ab modeling and Ab–Ag docking methods, but others could be employed; given sufficient data points for different targets, the impacts of these choices could be quantitatively assessed. The complete set of thousands of docking models, instead of cluster representatives, could be considered, thereby potentially leveraging redundancy in weighting region importance. As discussed above regarding gD, accounting for protein flexibility (both Ag and Ab) may change the models substantially in some cases. Docking scores/ranks could be taken into account in order to determine the most important regions. In addition to physically-based docking methods, additional data-driven epitope prediction methods could be used to identify putative antigenic regions [65,66,67].Dock binning. The same issues with clustering for experimental binning hold here and the same potential solutions can be explored. Machine learning methods could be incorporated in order to train models to integrate the upstream predictions and experimental data, e.g., in a consensus or weighted fashion. New data-driven models that directly seek to predict competition/bins (instead of general-purpose epitope prediction) could be developed.Experimental epitope probing. As presented, mutations are selected to assess the predicted antigenic hot spots and deconvolve which hot spots are associated with which Abs. While this has been done before based simply on maximizing binding disruption according to the models [9,61], more refined metrics and optimization techniques could be developed, e.g., using an information-theoretic approach to maximize what is learned about the communities and hotspots for a given experimental “budget” (e.g., maximum-relevance, minimum-redundancy) [68].Revised dock bins and antigenic hotspots. We employed the approach of docking with affinity/repulsion in order to focus docking according to the experimental data. Docking models could be subsequently refined (e.g., via energy minimization and conformational sampling) and scored not only for physical modeling, but also for the consistency of the experimental data with the predicted effects of the mutations [69].


## 4. Materials and Methods

All reagents were previously described [46,49,50,70,71].

### 4.1. Antibodies

Monoclonal Abs (mAbs) were used throughout. The monoclonal antibodies have been created by the Cohen/Eisenberg group at the University of Pennsylvania. The following anti-gD mAbs were previously published: 1D3 [49,72,73,74]; DL6, DL11, DL15 [51,71,74,75,76,77,78]; E317 [51]; MC1, MC2, MC4, MC5, MC8, MC9, MC10, MC14, MC15, MC23 [71,79]; A18 [80]; AP7, LP2 [81]; HD1, HD2, HD3, H162, H170, H193 [82,83]; 11S, 12S, 45S, 77S, 97S, 106S, 108S, 110S [73,84]; BD78, BD80 [71,85]; and the human mAb VID [86,87,88], DL15, A18, HD3, H162, H193, 77S, 97S, 106S, 108S and 3D5, 4E3E, 4G4D, and 11B3AG [46,70].

Sentinel Abs were selected based on past studies as most representative of their respective communities. Sentinel Abs were sequenced (Genscript, Piscataway, NJ, U.S.) from mAb clone for computational studies.

### 4.2. Proteins

Proteins were previously described [70]. Again, these constructs existed before this study and were generated by the Cohen/Eisenberg group. HSV type-1 and type-2 gD, truncated to 285 or 306 residues, were harvested from Sf9 cells infected with baculovirus. Protein was then purified using a DL6 immunosorbent column [70,71,89]. Additional proteins used in this study include: C-terminal truncations 250t, 260t, 275t, and 316t [50,54,70]; deletion mutant Δ(222–224) [50]; point mutants Y38A, V231W, and W294A [54,90,91]; and insertion mutants ins34, ins126, and ins243 [46,77].

### 4.3. Epitope Binning

Binning experiments were previously described [3,46]. Briefly, a 48-spot microarray of amine-coupled Abs on a CMD200M sensor prism (Xantec GmbH, Kevelaer, Germany) was printed using Continuous Flow Microspotter (CFM, Carterra, Salt Lake City, UT, U.S.). The array was loaded into the MX96 SPR instrument (Ibis). Data were processed using software from Ibis and Carterra as described previously [3,46].

### 4.4. Protein Modeling and Docking Prediction

The crystal structure of gD2 (285 truncation) was taken from PDB id 2C36 [54,55,56] and Schrodinger BioLuminate was used to homology model the unstructured regions for missing residues including 257–267. Homology models of the Ab Fvs were also constructed using Schrodinger BioLuminate. For E317, the homology modeling was prevented from using the available crystal structure (PDB id 3W9D); instead, the model was based on the anti-HIV neutralizing Ab 4E10 (PDB id 4M62) as the framework template, yielding a model RMSD value of 1.43 Å to the crystal structure.

Docking models were generated for each Ab model against the gD2 model using the Piper algorithm within Schrodinger BioLuminate [57]. This generated 15–30 different docking models for each sentinel Ab. We note that this method performs rigid docking, not accounting for potential Ab and Ag flexibility.

### 4.5. Dock Binning

Dock binning takes as input a set of Abs and a set of docking models for each Ab against the same Ag. It assesses docking models for the extent to which they structurally overlap or “compete”, and then clusters them based on their profiles of competition against each other in a manner analogous to experimental epitope binning. Finally, it constructs an antigenic heatmap on the Ag surface, highlighting Ag residues according to the frequency with which they contact Ab residues in the docking models comprising a cluster. We now detail these steps.

#### 4.5.1. Competition

Competition between a pair of docking models was assessed with three different scores [21] based on residue-level distances and biophysical interactions across the Ab–Ag interface:

Common interaction metric: the number of Ab–Ag atomic interactions such that one Ag atom has a common interaction with an Ab atom on each Ab. This was computed based on the Interaction Table in BioLuminate, which includes hydrogen bonds, salt bridges, pi stacking, disulfide bonds, and van der Waal interactions.

Cα metric: the number of Cα residues in one Ab within a fixed distance (here 10 Å) from those in the other.

Centroid metric: the distance between the heavy-atom centroids of the two Abs.

Results shown were based on the common interaction metric; results from the other metrics differed only minimally in terms of clustering results.

#### 4.5.2. Clustering

Competition scores were collected into a symmetric matrix, with each row and column representing a docking model and each cell containing the score for the associated pair of models. In this matrix, the row/column for a docking model collects its competition score against each docking model; we call this its “competition score profile”. A docking model heatmap was generated based on this matrix (schematically illustrated in Figure 3, with binary scores). The models were hierarchically clustered based on their competition score profiles, here using NBclust. The resulting dendrogram was partitioned to define clusters, also known as communities [92,93,94]. Carterra Epitope Binning software was used for dendrogram and community plot generation [46].

#### 4.5.3. Antigenic Heatmap

The notion of an antigenic heatmap was developed here to visually represent where the different docking model communities were located on the Ag, i.e., with which Ag residues the Abs in a community interacted the most. Each Ag residue was assigned to the docking model community for which the associated docking models had the most interactions according to the interaction tables generated by BioLuminate (see “common interaction metric” above). The “hotness” of the Ag residue was computed as the number of such interactions, normalized by the size of the community, and the heatmap shade was set accordingly.

In order to compare an antigenic heatmap “footprint” with an experimental epitope region, corresponding sets of residues were identified—those Ag residues comprising a community in the antigenic heatmap and those from the experimental epitope. The centroid of each set of residues was computed and the centroid distances measured, both using Schrodinger PyMOL. In addition, the sets of residues were directly compared for membership to identify how many residues were in common.

## Figures and Tables

**Figure 1 molecules-25-03659-f001:**
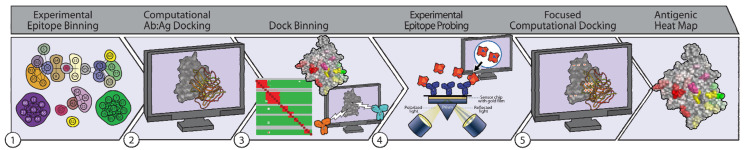
Schematic overview of our method for localizing Ab-specific epitopes by integrated experimental and computational analysis of binding competition. (Step 1) Experimental binning identifies communities of Abs that compete with each other for Ag binding. This grouping allows subsequent analyses to be focused on one or a few representatives from each community, reducing the effort required. (Step 2) For each representative Ab, a homology model of its Fv structure is constructed from its sequence, and Ab:Ag docking models are generated from the Ab model and the Ag structure or homology model. (Step 3) The docking models are computationally clustered, with this dock binning process analogous to experimental epitope binning in identifying patterns of competition. Since the competition is in terms of structural models of Ab:Ag binding, the identified communities correspond to general antigenic regions, and thereby map out potential binding regions on the Ag, summarized across the whole Ab panel as an antigenic heatmap. (Step 4) Experimental data is collected to probe the hypothesized epitope regions, e.g., using site-directed mutagenesis, chimeragenesis, peptide binding, or selection of alternative natural variants, in order to alter a putative epitope and evaluate effects on experimental competition or binding. (Step 5) The experimental data is used to focus docking, redefining bins, better characterizing competition, and better localizing epitope binding regions. This ultimately results in an antigenic heatmap localizing putative binding regions of the different Abs (illustrated by colored patches on the Ag surface).

**Figure 2 molecules-25-03659-f002:**
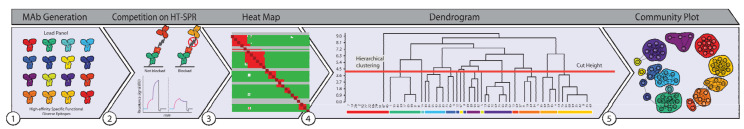
Schematic overview of experimental epitope binning, identifying cross-competition between Abs for binding an Ag. Sensorgrams are assessed for blocking, inferring whether or not two Abs block each other from binding in a competition assay and thus are assumed to bind the same epitopic region on the Ag. The blocking data is collected in a heatmap that indicates whether or not each Ab pair blocks (red blocked, green sandwiched). The heatmap is processed with hierarchal clustering algorithms based on similarity in blocking profiles, thereby generating a dendrogram. Finally, cutting the dendrogram separates Abs into clusters represented in a community network, which has nodes for the Abs and edges indicating which pairs block each other.

**Figure 3 molecules-25-03659-f003:**
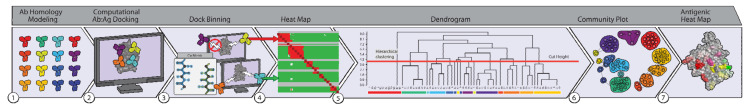
Schematic overview of “dock binning”, computational analysis of cross-competition of docking models of Abs against an Ag. Homology models of the Abs are computationally docked against a crystal structure or homology model of the Ag. Docking models are evaluated for “competition” and the extent of competition represented in a heatmap. The competition profiles are subsequently used to cluster docking models, with a community network representing the models (nodes), their competition (edges), and the identified clusters (groups of nodes). In contrast to experimental binning, dock binning is based on structural analysis and thus provides insights into where on the Ag the Abs might be interacting. The antigenic heatmap highlights the most popular Ag residues across the docking models; these are thus most generally informative for subsequent experimentally probing.

**Figure 4 molecules-25-03659-f004:**
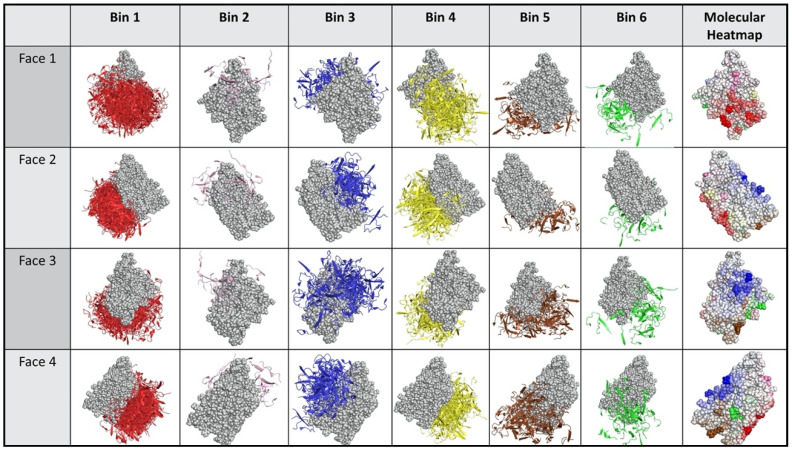
Binned docking models for Abs vs. gD2. Clusters of docking models are arranged in a table format, with each column a different cluster and each row a different perspective. Face #1 is the nectin binding face. Faces #2–4 are rotated by 90, 180, and 270 degrees around the *y*-axis. The rightmost column is an antigenic heat map (see also Figure 6) where residues are colored by community and shaded so that residues with more Ab interactions are colored more darkly.

**Figure 5 molecules-25-03659-f005:**
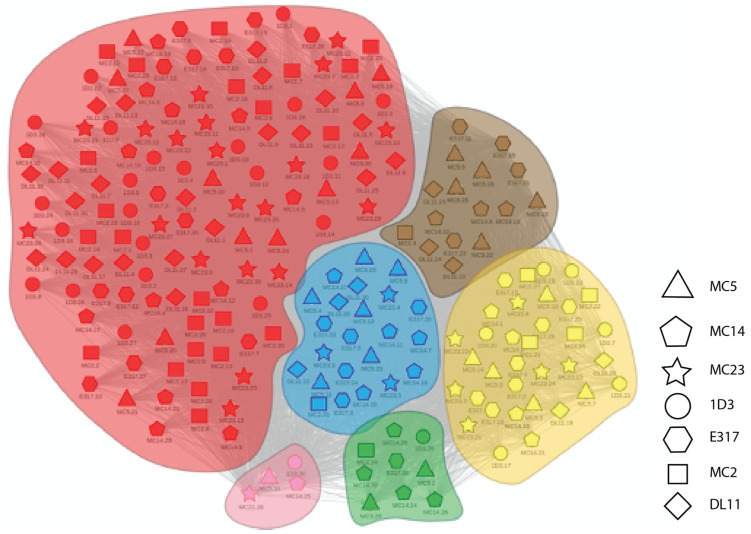
Community map of docking models for Abs vs. gD2. Nodes represent docking models, with different symbols for the different Abs. Edges indicates structural competition between pairs of docking models. Colors and background shading indicate community membership according to the partitioning of a dendrogram. The shape of each marker indicates the antibody in the dock (i.e. each shape is a single antibody).

**Figure 6 molecules-25-03659-f006:**
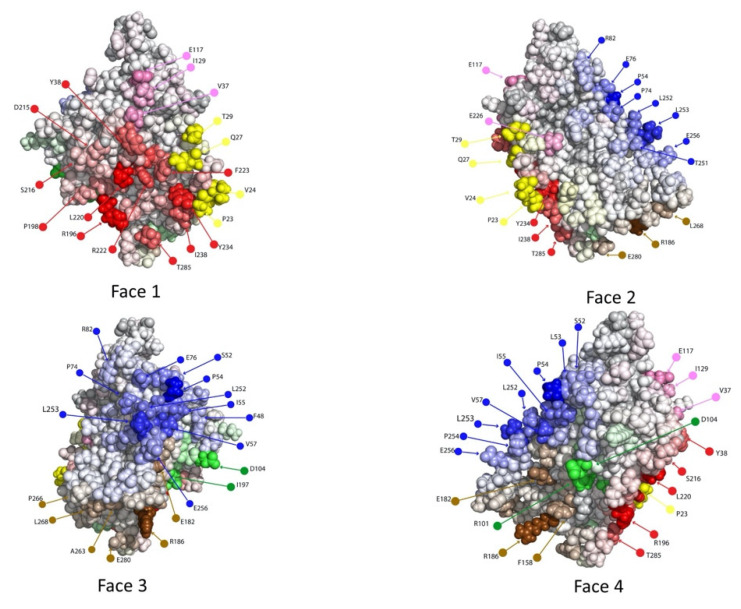
Antigenic heat map for Abs vs. gD2. Each residue is colored according to the Ab community (Figure 5) with which it has the most interactions and shaded so that residues with more interactions are colored more darkly. Face #1 is the nectin binding face. Faces #2–4 are rotated by 90, 180, and 270 degrees around the *y*-axis.

**Figure 7 molecules-25-03659-f007:**
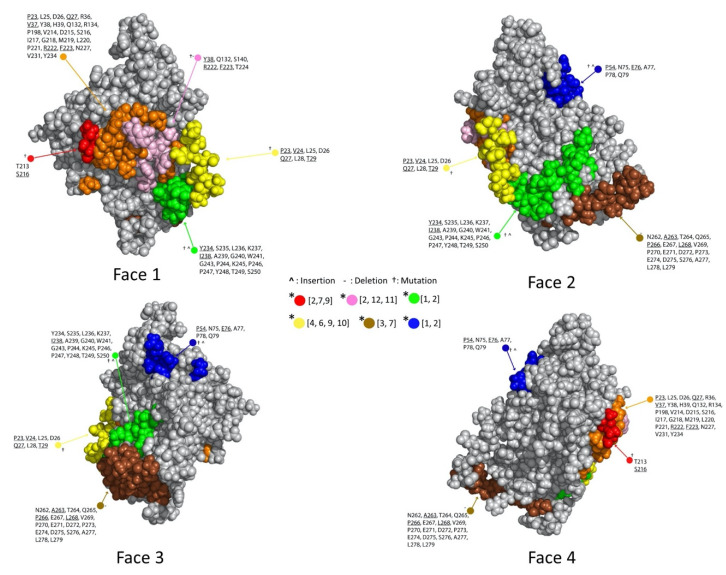
Experimentally probed residues. Residues with associated experimental binding data are labeled; bracketed numbers in the legend indicate primary references in the bibliography. The surface is colored by associated Abs (Appendix A), rather than dock binning community. The residues that were also highlighted by dock binning are underlined; this data serves as probes of those particular predicted antigenic hotspots.

**Figure 8 molecules-25-03659-f008:**
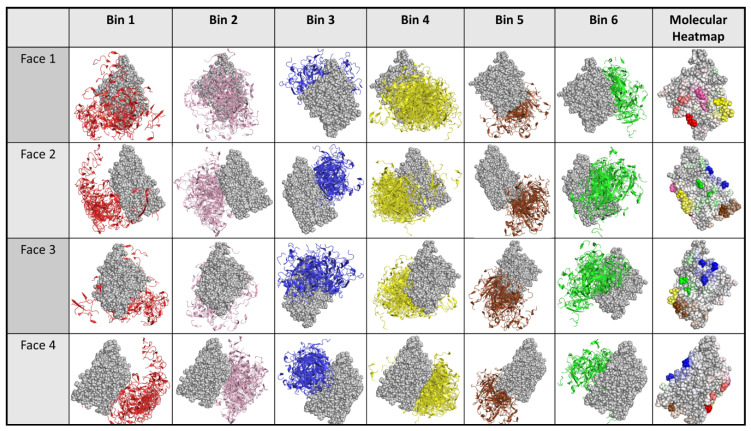
Binned experimentally-focused docking models for Abs vs. gD2. The same representation as Figure 4, but based on docking models that were focused according to experimental data.

**Figure 9 molecules-25-03659-f009:**
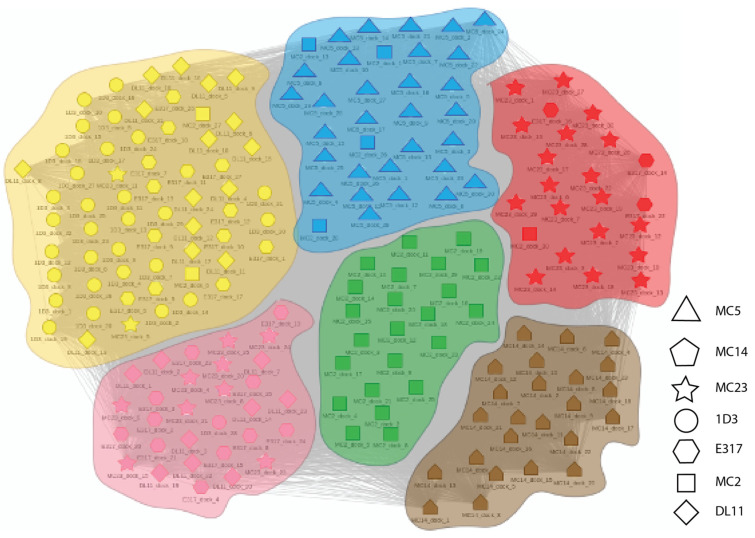
Community map of experimentally-focused docking models for Abs vs. gD2. The same representation as Figure 5, but based on docking models that were focused according to experimental data. Note the relative homogeneity of Abs (different symbols) within each community, contrasting with Figure 5. The shape of each marker indicates the antibody in the dock (i.e., each shape is a single antibody).

**Figure 10 molecules-25-03659-f010:**
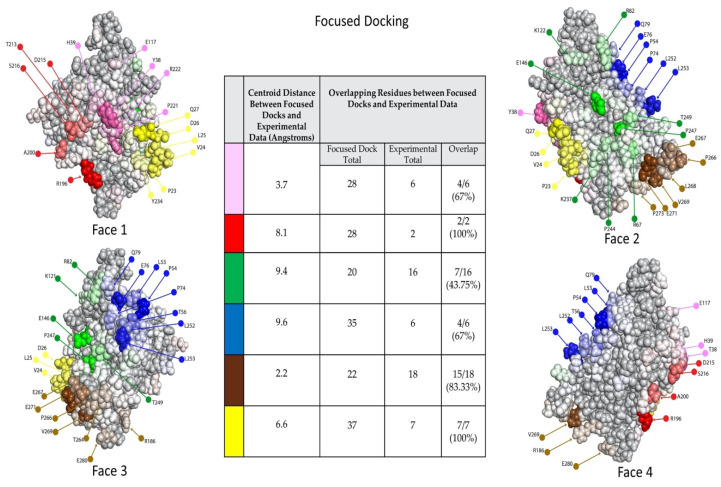
Antigenic heat map for Abs vs. gD based on eperimentally-focused docking models. The same representation as Figure 6, but based on docking models that were focused according to experimental data. The resulting hot spots can be interpreted as likely epitope regions of the associated antibody/-ies in the bin. Relative.

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
