# Peer review of "Characterizing Epitope Binding Regions of Entire Antibody Panels by Combining Experimental and Computational Analysis of Antibody: Antigen Binding Competition"

_molecules, 2020, doi:10.3390/molecules25163659_

Round 1

Reviewer 1 Report

In this paper, the research team employed a combination of SPR competition data and virtual docking predictions to improve prediction of binding locations for numerous Abs. This data was then exploited in the design of specific mutagenesis studies that experimentally tested various virtual docking predictions and led to far more robust predictions as to the epitopes recognized by many of the Abs. Through these studies, the authors successfully demonstrate that this new approach could be an important tool for well-resourced researchers seeking robust data on the epitopes targeted by large panels of diverse Abs to a specific target.

What is less clear is how useful this approach would be under less ideal circumstances such as when the Ab panels are less diverse, the target is more difficult to express in its native conformation, or when particularly immunodominant epitopes are present. Even under ideal circumstances, the results are models that must be viewed as inconclusive until specific structural data validates the predictions. Nevertheless, this approach seems likely to become increasingly powerful, particularly in combination with new advances in machine learning technology and improving docking prediction software, after which it will likely become more broadly applicable to many other antibody discovery projects.

Specific comments:

  1. Authors employed SPR for Ab binding competition and Fig. 2 shows ideal outcomes for no competition or for complete competition. It seems unlikely that all SPR outcomes were so clear. How did the authors handle partial or aberrant competition results compared to controls? Aren’t these potentially informative and relevant, perhaps beneficial, for the binning predictions?
  2. As a person involved in antibody discovery, yet not trained as a structural biologist and unfamiliar with the software employed, it was nonetheless possible for this reviewer to appreciate the value of the approach and to generally follow the results. On the other hand, it was very difficult to specifically follow the detailed research process using the information provided. Some figures are difficult to follow and offer limited help in the legends. For example:
    1. Fig. 1- text mentions 46 Abs studied yet shows over 60 in dendrogram, clarify and perhaps indicate which are the sentinel Abs
    2. Figs 4 and 9- improve resolution and explain significance of colors, shapes, positions on various faces, shape identifiers, etc
    3. Fig. 7- colors don’t always match the key, clarify bracketed numbers in key
    4. Fig. 9- legend incorrectly refers to Fig 5 instead of Fig. 4.
  3. To more easily understand the extent of the experimental mutagenesis performed to achieve the improved epitope predictions, the text should summarize the specific numbers and types of mutations employed to achieve the focused docking reported in Fig. 10. It appears that about 55 different mutations were studied, which would be very difficult for low-resource labs or those working on difficult to express targets. Authors should also explain how they arrived at the number of mutations selected, and how they decided to employ insertions, deletions or mutations.
  4. The methods section seems very superficial considering the extensive work underpinning this report. While it is fine to reference detailed methods, some brief descriptions would save readers from having to dig into references just to understand basic methods employed. Almost nothing is mentioned regarding methods for Ab discovery, site-specific mutagenesis and protein expression and purification.

Author Response

Attached is a response to the reviewer.

Reviewer 2 Report

This paper is poor, both in terms of scientific rigour (logical flow, rationale and sufficient experimental detail) and also in terms of presentation and attention to detail.  

The paper mixes Materials and Methods with the Results (much of the text currently in  Results should move to Materials and Methods).  The Methods are not described in sufficient detail to allow others to reproduce or test the work.  The single test system for the method "gD2" is hardly explained or justified - why was this system chosen?  What is the "285 residue truncation of gD2" and why was this one used? Why only one test case?  There is no substantive analysis of the validity or reliability of the docking and only one docking algorithm is used.  There is no discussion/exploration of the limitations of the methods - for example, I assume that flexibility of the Ag was not considered in the docking? Further, the authors claim that "Ab homology modeling is generally very high quality" - what does "high quality" mean in this context?  Precision in atomic position (which homology modelling provides) is not the same as accuracy!  

The results are poorly presented.  Many figures are mere schematics and do not present data that can be interrogated by the reader (i.e. Figures 1-4,9). How is the reader mean to explore Figure 9? Why was no clear example of an 'antigenic heatmap' presented and explained? Worse, the captions for Figures 4, 5 and 6 are all mixed up (incorrect captions for each figure) and are completely nonsensical as presented.  The subsections headings are far too long and very peculiar (e.g. "While binning is still limited, further computational advances will improve its accuracy and informativeness.

"). Indeed, there was a very inadequate  final check of this manuscript before submission. There are numerous grammatical errors, mixed fonts (obvious from sloppy cut-and-paste) incomplete sentences, missing words  (e.g. "the more there is for interaction") and some bizarre terminology (e.g. "we scale up this general idea" - how do you scale up an idea? - "leverages the intuition introduced above" -??).

Finally, there are references missing required details such as page numbers, journal numbers etc. (e.g. Refs 20,21,31) and the citations are messy - missing the required spacing, appearing both before and after punctuation, with and without spaces.

The article has 8 self-citations - which is high (refs [5,20,21,30,31,45,47,73]). Further, these references are used to support the broad claims of the authors - this kind of supporting citation for a primary claim really should be to others work. Examples below "One of the areas for improvement in Ab engineering is the characterization of the binding interactions between an Ab and its specific Ag, as defined by the epitope:paratope interface [*20,21*].” Surely others have done work in this area? “Foremost amongst emerging techniques for high-throughput epitope characterization is epitope binning as it merges the speed and functional-site identification capabilities demanded by the biopharmaceutical industry [3,4,20,31] " “This approach can enable the pipeline to be populated with Abs that are diverse epitopically and consequently functionally [3,4,20,31]. “ However, these are minor points. My other comments on the very poor quality of this article are more substantive. I would not have said anything about this in the report, but there was a checkbox for it.

Reviewer 3 Report

The manuscript describes an experimental and computational pipeline which can be used to derive potential epitope sites using a combination of experimental and computational docking-based binning. While the novelty of this approach is generally limited, the authors do provide some interesting strategies on how to analyze and reduce large antibody panels early in therapeutic Ab discovery campaigns. I do recommend this manuscript for publication after addressing the following points below.

Page 3, lines 4 and 5: No need to cite the same 4 references twice here.

Page 4, first paragraph of results: change sentence to either “fail to prevent…” or “do not prevent”

Page 4, last sentence: The publication referenced with regards to the E317 Fab – gD structure is not correct but rather describes the structure of HSV-2 gD in free form and in a complex with nectin-1. Please cite the correct publication here (Lee et al. Acta Crystallogr D Biol Crystallogr. 2013 Oct;69(Pt 10):1935-45). It would also be helpful if the authors could provide the pdb code of this structure (3W9E).

Page 4, line 6: I fail to understand why reference 50 is cited here. Please check and remove.

Figures 2 & 3: I think it would be more beneficial for the reader if figures 2 and 3 were combined into a single figure. This would facilitate comparison of individual steps during experimental and dock binning.

Figures 4 & 5:

  • Figures are in the incorrect order. Figure caption 4 is assigned to figure 5 and vice versa.
  • Current figure 4 caption mentions face #5 and face #6 as views from above and below. However, there are no such views in the corresponding figure. The same is true for figure 6 where face 5 and 6 are missing as well.
  • Please correct “Edges indicates…” in current figure 5 caption and further explain meaning of the sentence. Should this be “Overlapping edges indicate…”?. Also, please add a legend to figure 4 indicating mAb identity of each symbol.

Figure 10: The last sentence in the figure caption doesn’t make much sense and needs to be changed.

Page 13: Check interpunction in the first sentence of the Experimental epitope binning section.

Discussion:

While the presented summary of potential avenues in terms of computational developments is certainly an interesting addition to the manuscript, I would like to see the discussion extended towards the actual results of the paper. Are there any common features of residues that where not found or additionally found in the focused dockings as compared to the experimental studies? Could this be explained by current limitations of the docking algorithms? Without preexisting knowledge about potentially neutralizing candidates within the mAb panels, how likely would it be to identify therapeutically promising candidates from these results?

Check format of references 20, 21, 29, 31, 35, 46, 60, 100

Round 2

Reviewer 2 Report

This second version version of the paper is improved, but mostly with regards to some extra explanation in the text and the figures appearing in the right order, correctly labelled.  It is a concern that the authors do not systematically address some key substantive issues I raised in the last version.  It is standard procedure in a review to list each one of the issues and address them individually - not to do a blanket response to an entire group of issues (and thereby ignore some of them). The issues raised in the last version that have not really been addressed follow.

There is no substantive analysis of the validity or reliability of the docking and only one docking algorithm is used. This is not a case of whether you are defining a benchmark (??)  - all scientific methods need an estimate of reliability or error.  The authors claim that "While not at atomic resolution, this approach allows for group-level identification of functionally related monoclonal antibodies (i.e., communities) and identification of their general binding regions on the antigen."  This claim is completely reliant on the accuracy of the docking algorithm used.  Also, it is an interesting and pertinent question whether the results vary significantly with different docking algorithms. This should be address and not glossed over.

There is no discussion/exploration of the limitations of the methods - for example, I assume that flexibility of the Ag was not considered in the docking? This is not addressed at all by the authors/

Why was no clear example of an 'antigenic heatmap' presented and explained?The authors now claim that Figure 6 and 10 show heatmaps. However, these figures do not have heatmaps - they have molecular structures with some residues coloured.  A Heatmap is a 2-dimensional graph. The paper seems to imply that they do produce such heatmaps (symbols of which are shown in Figures 2 and 3) - but there is still no explanation of these of what exactly they show, or even clear a clear example of one. Worse, the paper conflates the actual map with a colour mapping of the 3D structure.  At minimum, the incorrect use of the term "heatmap" should be removed.

The abstract has not been altered at all to reflect the changes in the paper and is  (still) too long as per the journal's Instructions to Authors. The paper would also benefit from an additional edit to remove the remaining grammatical errors in the document. 

Author Response

This second version version of the paper is improved, but mostly with regards to some extra explanation in the text and the figures appearing in the right order, correctly labelled.  It is a concern that the authors do not systematically address some key substantive issues I raised in the last version.  It is standard procedure in a review to list each one of the issues and address them individually - not to do a blanket response to an entire group of issues (and thereby ignore some of them). The issues raised in the last version that have not really been addressed follow.

We apologize for any oversight. We believed that the substantial changes in our explanations, throughout the paper (introduction, methods, and results), were central to addressing the concerns raised by the three reviewers. Furthermore, we believed that all substantive issues had been systematically addressed in the manuscript and/or responded to in the letter. As is common, our letter responded separately to each paragraph of comments, assuming that the reviewer had grouped the questions into a paragraph on purpose (e.g., the set of target-related questions raised by this reviewer), and that we should not arbitrarily break them up and risk disrupting coherence. Likewise, we grouped our response paragraphs topically to ensure focused coverage of each subset of related concerns.

We hope that the following response and associated revisions in the manuscript will serve to clarify any remaining concerns.

There is no substantive analysis of the validity or reliability of the docking and only one docking algorithm is used. This is not a case of whether you are defining a benchmark (??)  - all scientific methods need an estimate of reliability or error.  The authors claim that "While not at atomic resolution, this approach allows for group-level identification of functionally related monoclonal antibodies (i.e., communities) and identification of their general binding regions on the antigen."  This claim is completely reliant on the accuracy of the docking algorithm used.  Also, it is an interesting and pertinent question whether the results vary significantly with different docking algorithms. This should be address and not glossed over.

We acknowledge that the reliability of all parts of the pipeline – antibody modeling, docking, clustering, etc. -- are critical to the utility of the method. But this is exactly why benchmarks exist, and why we rely on them. Rather than evaluating various methodological and implementational choices for each part of the pipeline, which in theory would have to be considered not just independently but combinatorially, we refer to previous evaluations to select high-quality representative approaches. For example, there exists plenty of literature and even a regular competition evaluating the reliability of docking methods. We then count on the chosen method to work in this scenario to the same degree of quality as the benchmarks suggest that it will. The same is true of antibody modeling, clustering, etc., in our work as well as in other methods that leverage such pieces – progress is made by assuming the trends observed in benchmarks will hold in further studies, and thus the consideration of one choice for each part of the pipeline is sufficient to establish the approach.

We are not sure why the reviewer inserted ‘(??)’ after ‘benchmark’; perhaps he/she is not familiar with this practice, which is absolutely necessary in engineering and computer science, so as to avoid having to test everything from the ground up every time (again, in combination). The literature is full of papers that use one representative, high-quality (from benchmarks) implementation as a component in another approach. Stepping along our pipeline and even just considering some of the references in our bibliography:

  • Ab homology modeling is a key first step for SnugDock [40], and the method is quite generic to the actual model generation, but (quite naturally) they only test RosettaAntibody models.
  • Docking is central to the epitope determination method presented in [42], but they only consider Piper.
  • In their seminal paper, Eisen and colleagues [63] point out that many clustering variations are possible, yet for clarity they prefer to present only one form of hierarchical clustering with one metric and one linkage.

We selected Schroedinger’s Bioluminate for its previously established reliability and validity. Similarly the other implementations we selected were based for demonstrated effectiveness, allowing us to reliably build on them.

We agree that the impact of the docking method on the results is pertinent, and in theory it may be interesting. However, in our experience here and in other studies (see e.g., Hua et al., eLife 2017), when we preliminarily tested a few different methods, the differences did not manifest any systematic effects of interest. Furthermore, with the relatively small set being evaluated here, any such effects could not be quantified to any degree. Thus we decided that presenting results from multiple methods each step along the pipeline would only muddy the story, and would best be done in future studies with enough data to establish quantitative comparisons. That is again the value of benchmarks -- unit testing of the steps along the pipeline, since there is enough data for a single step to enable quantitative comparison of methods for that step, even though there isn’t enough data regarding the same coherent system to enable integrated testing of all choices. We have elaborated the relevant discussion to that effect.

Finally, we would argue that Fig. 10 establishes our claim “this approach allows for group-level identification of functionally related monoclonal antibodies (i.e., communities) and identification of their general binding regions on the antigen”. It does not establish a claim that any choice of method for each step along the pipeline will work, or that the choices we made will work on every system, but it does prove the principle that these choices worked for this system, and thus the approach has merits.

There is no discussion/exploration of the limitations of the methods - for example, I assume that flexibility of the Ag was not considered in the docking? This is not addressed at all by the authors/

We refer the reviewer to the Discussion section, where we indeed discuss and explore the limitations of the methods. The impact of Ag flexibility is discussed in the last paragraph in section 3.1. Other limitations and potential extensions, including for docking and clustering, are raised in section 3.2. As discussed above, we elaborated that a bit to point to the reviewer’s concern as one that could be addressed with sufficient data.

Why was no clear example of an 'antigenic heatmap' presented and explained?The authors now claim that Figure 6 and 10 show heatmaps. However, these figures do not have heatmaps - they have molecular structures with some residues coloured.  A Heatmap is a 2-dimensional graph. The paper seems to imply that they do produce such heatmaps (symbols of which are shown in Figures 2 and 3) - but there is still no explanation of these of what exactly they show, or even clear a clear example of one. Worse, the paper conflates the actual map with a colour mapping of the 3D structure.  At minimum, the incorrect use of the term "heatmap" should be removed.

We reiterate that Figures 6 and 10 show what we term ‘antigenic heatmaps’. The reviewer seems to take an overly restricted view on how a heatmap can be displayed. We refer for example to the Wikipedia entry (https://en.wikipedia.org/wiki/Heat_map), which shows a heatmap on the surface of the globe – a visualization quite analogous to our antigenic heatmaps in that both the surface of the globe and the surface of a protein are, mathematically, 2D surfaces encompassing 3D bodies. Much as flattening a globe into a rectangular map leads to distortions and discontinuities (what’s on the right edge is actually adjacent to what’s on the left edge), flattening a protein surface would not provide an accurate depiction of the antigenic regions. Thus we prefer to leave our heatmap embedded in 3D to clearly illustrate the contiguous parts of the surface most likely to be antigenic for different antibodies.

The figure legend gives an intuitive description of how the heatmap is defined, and the main text near the figure further elaborates. We also refer the reviewer to the methods section, where we included a subsection describing in even more detail the construction of an antigenic heatmap. The “heat” is defined in terms of frequency of docking model contacts, which we trust that the reviewer will agree is appropriate for heatmapping, and it is visualized on the antigen’s 2D surface, displayed using color gradients that we likewise trust the reviewer will agree are suitable representations of the “heat”. We maintain that these choices are entirely consistent with many usages of heatmaps, and our representation is in fact a correct use of the term “heatmap”.

The abstract has not been altered at all to reflect the changes in the paper and is  (still) too long as per the journal's Instructions to Authors.

None of the changes in the paper affect the high-level summary in the abstract, and the abstract is consistent with the main body. If the editor feels that it is necessary to chop 25 words, we will do so, but we believe that the abstract as currently written nicely summarizes the presented in the manuscript.

The paper would also benefit from an additional edit to remove the remaining grammatical errors in the document. 

We have taken another editorial pass.